# Epidermal growth factor receptor (EGFR) and vascular endothelial growth factor A (VEGF-A) expressions in Ethiopian female breast cancer and their association with histopathologic features

**Sisay Addisu**[1]*, **Abebe Bekele**[2¤], **Daniel Seifu**[1¤], **Mathewos Assefa**[3], **Tufa Gemechu**[4], **Mark J. Hoenerhoff**[5], **Sofia D. Merajver**[6]

1 Department of Biochemistry, School of Medicine, College of Health Sciences, Addis Ababa University, Addis Ababa, Ethiopia, 2 Department of Surgery, School of Medicine, College of Health Sciences, Addis Ababa University, Addis Ababa, Ethiopia, 3 Department of Internal Medicine, School of Medicine, College of Health Sciences, Addis Ababa University, Addis Ababa, Ethiopia, 4 Department of Pathology, School of Medicine, College of Health Sciences, Addis Ababa University, Addis Ababa, Ethiopia, 5 In Vivo Animal Core, Unit for Laboratory Animal Medicine, University of Michigan Medical School, Ann Arbor, Michigan, United States of America, 6 Division of Hematology and Oncology, Department of Internal Medicine, University of Michigan Rogel Cancer Center, Ann Arbor, Michigan, United States of America

¤ Current address: School of Medicine, University of Global Health Equity, Kigali, Rwanda
* sisay.addisu@aau.edu.et

**Data Availability Statement:** All relevant data are within the manuscript.

## Abstract

### Background

Epidermal growth factor receptor (EGFR) and vascular endothelial growth factor receptor (VEGF) play important role in breast tumor growth, invasion, metastasis, patient survival and drug resistance. The aim of this study was to evaluate the protein expression status of EGFR and VEGF-A, as well as their association with hormone receptor status and histo-pathological characteristics in the invasive type of female breast cancer among Ethiopians.

### Method

The primary breast tumor tissues were obtained from 85 Ethiopian invasive breast cancer cases that underwent modified radical mastectomy (MRM) from June 2014 to June 2015. Their FFPE blocks were analyzed for EGFR and VEGF protein expressions using immuno-histochemical techniques. The expressions were also correlated with histopathologic features.

### Result

Epidermal growth factor receptor over-expression was observed in 22% of the tumor samples. VEGF-A expression was negative in 13.41%, low in 63.41%, moderate in 20.73%, and high in 2.44%. EGFR expression, but not VEGF-A, showed a significant inverse correlation with both estrogen receptor (ER) ($P = 0.01$) and progesterone receptor (PR) statuses ($P =$

**Funding:** This work is funded in part by Addis Ababa University and International Reproductive Health Training (CIRHT) program at the University of Michigan. No additional external funding was received for this study. The funder had no role in study design, data collection and analysis, decision to publish, or preparation of the manuscript.

**Competing interests:** Authors have declared that no competing interests exist.

0.04). EGFR and VEGF expressions did not show significant association with tumor size, grade, lymph node status or age at diagnosis.

## Conclusion

Epidermal growth factor receptor expression was most likely associated with ER and PR negative tumors. Assessments of multiple molecular markers aid to understand the biological behavior of the disease in Ethiopian population. It might also help to predict which group of patients might get more benefit from the selected treatment strategies and which are not.

## Introduction

Breast cancer is a complex disease that exhibits variations in clinical, morphological and biological characteristics [1]. Currently, five treatment options are used to address breast cancer: surgery, radiotherapy, chemotherapy, hormonal therapy and immunotherapy. The selection of treatments is based on the grade, stage and molecular subtype of the cancer [2]. Surgery, chemotherapy, and radiotherapy are commonly used for all types of breast cancer, while endocrine therapies are employed for hormone receptor-positive cases and anti-HER2 therapy for HER2+ cases, targeting specific molecular subtypes [2, 3]. Due to its heterogeneous nature and lack of molecular targets, treatment of triple-negative breast cancer (TNBC) patients mainly depends on surgery, chemotherapy and radiotherapy with supplementation of limited targeted therapies. These include Poly(ADP-ribose)polymerase (PARP) inhibitors, immune checkpoint inhibitors and antibody-drug conjugate therapies [4–6]. Despite recent advancements in targeted breast cancer treatment, challenges persist in the form of therapeutic resistance, disease recurrence, and metastasis, influenced by other signaling molecules and pathways [7, 8]. Among other factors, the expression of VEGF and EGFR has been shown to play a role in therapeutic resistance against breast cancer treatment.

Epidermal growth factor receptor plays a significant role in carcinogenesis due to its oncogenic activity through various mechanisms [9, 10]. Previous studies have reported an association between EGFR expression and a poor response to tamoxifen treatment in breast cancer. The signaling cross-talk between ER and growth factor receptor pathways leads to resistance against hormonal therapy [11, 12]. EGFR expression is also associated with poor tumor differentiation, ER negativity, a higher mitotic index, and decreased survival [13]. Increased expression of EGFR is also associated with resistance to anti-HER2-targeted therapy in HER2 + breast cancer cases [14]. Anti-EGFR therapies may benefit individuals with EGFR overexpressing breast cancer and could serve as a predictive marker for EGFR targeted treatments. However, despite the numerous preclinical and clinical trials investigating anti-EGFR inhibitors in breast cancer, only two of them, Lapatinib and Neratinib, have been approved by the Federal Drug Administration (FDA). Lapatinib, an anti HER1/HER2 targeted drug, is approved for hormone positive and HER2-positive breast cancer. On the other hand, Neratinib, an anti HER1, HER2, and HER4 tyrosine-protein kinase inhibitor, is approved for HER2 positive breast cancer under specific conditions [15–17].

Vascular endothelial growth factor family members have been recognized to play fundamental roles by activating their receptors and have also been shown to be overexpressed in breast cancer [18, 19]. Among them, VEGF-A has been identified as the most potent angiogenic factor, inducing proliferation, and migration in endothelial cells. Additionally, it promotes tumor cell proliferation, survival and migration [18]. Higher expression of VEGF-A in

breast cancer cases has been significantly correlated with poor prognosis and shorter overall survival in some studies [20–22]. Additionally, VEGF expression has been associated with resistance to hormonal therapy in hormone-receptor positive breast cancer cases [23]. VEGF targeted therapies such as Bevacizumab and angiogenesis-related tyrosine kinase inhibitors were approved for the treatment of various malignancies [3]. However, the antitumor response to Bevacizumab in breast cancer patients was not as expected, leading to a revision of the FDA decision due to inconclusive results regarding progression-free and overall survival benefits. There is currently no approved VEGF-targeted therapy for the treatment of breast cancer [19, 23, 24].

Breast cancer is the most common type of cancer in Ethiopia [25]. However, there is limited information available about its molecular characteristics To the best of our knowledge, no study has reported on the status of VEGF and EGFR expression in Ethiopian female breast cancer. This study is the first of its kind to explore the expression of VEGF and EGFR, as well as their association with hormone receptors and clinico-pathological features in Ethiopian females with invasive ductal carcinoma of the breast.

## Materials and methods

### Ethical considerations

Ethical approval was obtained from the Institutional Review Board (IRB) of the College of Health Sciences at Addis Ababa University (Meeting No: 007/2015; Protocol number: 010/14/Bioch.), and the Federal Health Research Ethics Review Committee of the Ministry of Science and Technology in Ethiopia (Ref. No. 310/023/2015). A written informed consent statement was obtained from each study participant.

### Study participants and tumor samples

The present study involved female patients with clinically and pathologically confirmed invasive breast carcinoma. These patients underwent MRM, which included axillary lymph node dissection, between June 2014 and June 2015. The resected tissues were sent to two referral hospitals (St. Paul Millennium Medical College (SPMMC) and Menelik II Memorial Referral Hospital (MIIMRH)) for morphological analysis and FFPE preparation. Some FFPE blocks were excluded from the study due to reasons such as being undetectable or having non-invasive tumors. Ultimately, 85 female patients with invasive breast cancer who met the inclusion criteria were included in this study. Inclusion criteria were women aged 18 years old or older at the time of diagnosis, admitted for modified radical mastectomy, and clinically and histologically confirmed with invasive breast cancer. Females aged less than eighteen years old at the time of diagnosis, male breast cancer cases, females diagnosed only with fine needle aspiration cytology and non-invasive breast cancer cases were excluded from the study.

Due to inadequate clinical information such as clinical stage, menopausal status and family history, patient data were obtained from the standard histopathology requesting and reporting forms. The information gathered included age at diagnosis, tumor size, histologic type, tumor grade and lymph node status. Immunohistochemical biomarkers, including ER, PR, EGFR, and VEGF-A, were examined at the University of Michigan. Pathologic grading was based on the Nottingham combined histological grade [26]. Pathologic classifications of axillary lymph node involvement and tumor size were determined according to the American Joint Committee on Cancer (AJCC) definition [27].

## Immunohistochemical analysis

Immunohistochemical analysis was conducted using deparaffinized sections of FFPE tumor tissues. The sections were deparaffinized, rehydrated, and subjected to epitope retrieval with FLEX target retrieval solution (TRS), high pH. Epitope retrieval for EGFR staining was performed using Proteinase K ready-to-use reagent. Primary antibodies for ER (clone SP1, Rabbit monoclonal, Cell Marque, Rocklin, CA), PR (clone Y85, Rabbit monoclonal, Cell Marque, Rocklin, CA), EGFR (clone 31G7, Mouse monoclonal, Invitrogen, Camarillo, CA) and VEGF (A-20 SC-152, Rabbit polyclonal, Santa Cruz, CA) were applied using automated immunostainer (Dako Autostainer Link, Carpintaria, CA, USA). After applying specific monoclonal antibodies (for ER, PR and EGFR) and polyclonal antibody (VEGF), FLEX+ Mouse/Rabbit EnVision Systems (Agilent, Dako) were used for detection. The IHC slides were then scanned for ER, PR, EGFR and VEGF using the Aperio microscope slide scanning and scoring system (Aperio Technologies, Inc., Vista, CA).

## Immunohistochemical evaluation

The evaluation of the IHC stained slides of ER and PR was conducted using Allred's scoring system [28]. The percentage of positive tumor cells was graded on a scale of 0–5, and the intensity of labeling was graded on a scale of 0–3. The total score was calculated by adding the proportion and intensity scores, resulting in a range of 0–8 (0–2 labeled as negative and 3–8 labeled as positive).

The score for EGFR protein expression was reported based on previous publications [29, 30]. EGFR staining was considered positive when at least 10% of tumor cells exhibited strong membranous staining. Staining in normal adjacent tissues and stromal labeling was not considered as positive staining. For the evaluation of VEGF expression level, a semi-quantitative scoring system was used, which combined scores for the proportion of tumor cells stained positive and the intensity of the cytoplasmic or nuclear staining. This scoring system was previously described by Dhakal et al. [31] (see Table 1).

## Statistical analysis

We conducted statistical analysis of our data using SPSS 24.0 software (SPSS Inc., Chicago, IL, USA). Descriptive statistics were used to summarize the frequency distributions. The Pearson's chi-square test was employed to determine associations between categorical variables. Fischer's exact test was used as necessary. A two-tailed P value was of $< 0.05$ was considered statistically significant for the results.

# Results

## Patient and tumor characteristics

The age distribution of the patients ranged from 22 to 75 years (mean ± standard deviation (SD), 42.14 ± 11.96). The frequency percentages of the age groups for the 85 study participants and their tumor histopathologic characteristics are shown in Table 2.

**Table 1. Scoring system for VEGF-A protein expression.**

| % Positive Cells | Proportion Score | Intensity | Intensity Score |
|---|---|---|---|
| 0 | 0 | No staining of tumor cells | 0 |
| ≤10% of positive cells | 1 | Weak staining | 1 |
| 10–50% of positive tumor cells | 2 | Moderate staining | 2 |
| >50% of positive tumor cells | 3 | Strong staining | 3 |

**Table 2. Clinicopathological features.**

| Characteristics | Count | Percentage frequency (%) |
|---|---|---|
| **Age (years)** | | |
| Mean ± SD: 42.14 ± 11.96 | | |
| Range: 22–75 | | |
| **Age** | | |
| <50 | 60 | 70.6 |
| ≥50 | 25 | 29.4 |
| **Histological subtype** | | |
| Invasive Ductal Carcinoma | 74 | 89.2 |
| Invasive Lobular Carcinoma | 3 | 3.6 |
| Mixed ductal and Lobular Carcinoma | 3 | 3.6 |
| Mucinious carcinoma | 2 | 2.4 |
| Papillary carcinoma | 1 | 1.2 |
| Missing | 2 | |
| **Tumor Size** | | |
| T1 | 9 | 11.3 |
| T2 | 22 | 27.5 |
| T3 | 26 | 32.5 |
| T4 | 23 | 28.8 |
| Missing | 5 | |
| **Histological Grade** | | |
| G1 | 8 | 11.3 |
| G2 | 27 | 38.0 |
| G3 | 36 | 50.7 |
| Missing | 14 | |
| **Lymph Node Status** | | |
| N0 | 20 | 26.0 |
| N1 | 30 | 39.0 |
| N2 | 23 | 29.9 |
| N3 | 4 | 5.2 |
| Missing | 8 | |
| **ER status** | | |
| Positive | 62 | 73.8 |
| Negative | 22 | 26.2 |
| Missing | 1 | |
| **PR status** | | |
| Positive | 50 | 60.2 |
| Negative | 33 | 39.8 |
| Missing | 2 | |
| **EGFR status** | | |
| Positive | 18 | 21.7 |
| Negative | 65 | 78.3 |
| Missing | 2 | |
| **VEGF status** | | |
| High | 2 | 2.4 |
| Moderate | 17 | 20.5 |
| Low | 53 | 63.9 |
| Negative | 11 | 13.3 |
| Missing | 2 | |

Based on histologic subtype, 89% (74/83) of primary breast cancer samples were identified as IDC type. More than 60% (49/80) of our cancer samples belong to T3 and T4. Similarly, more than 70% (57/77) of the breast cancer tumors showed axillary lymph node metastasis. Among the histologically assessed tumor grades, 51% (36/71) belonged to poorly differentiated tumors (grade III) and 38% (27/71) were moderately differentiated (grade II) tumors. The lymph node status of eight cases was missed in the histopathology report.

Regarding the status of hormonal receptors, the majority of the study samples showed positive expressions for ER (73.8%) and PR (60.2%). Membranous EGFR staining was observed in varying percentages of tumor cells, ranging from 0 to 99%. Specifically, 17 cases (20.5%) had an expression of less than 10%, 6 cases (7.2%) had an expression between 10% and 25%, 1 case (1.2%) had an expression between 25% and 50%, 5 cases had an expression between 50% and 90%, 6 cases had an expression greater than 90%, and 48 cases (57.8%) showed no expression. Tumors with strong in more than 10% of the tumor cells were considered EGFR positive, which was observed in only 18 cases (21.7%).

Vascular endothelial growth factor A expression was calculated by multiplying the proportion and intensity scores with a range of 0 to 9. Scores from 0 to 3 were categorized as no∕low expression, 4 to 6 as moderate expression, and greater than 6 as high expression. In this study VEGF-A expression was observed as high in 2 (2.4%), moderate in 17 cases (20.5%) at moderate concentration, low in 53 cases (63.9%) at a low concentration, and negative in 11 cases (13.3%).

## Association of EGFR and VEGF-A with clinicopathological features

The correlations between the overexpression of EGFR and clinicopathological features are shown in Table 3. EGFR overexpression showed a significant association with ER (P = 0.01) and PR (p = 0.04). However, EGFR overexpression was not significantly associated with age, tumor size, histological grade (P = 0.16), and axillary lymph node involvement.

**Table 3. Association of EGFR overexpression with clinicopathological features.**

| Characteristics | EGFR | | |
|---|---|---|---|
| | Positive, N (%) | Negative, N (%) | p-Value |
| **Age** | | | |
| <50 | 14 (24.1) | 44 (75.9) | 0.41 |
| ≥50 | 4 (16.0) | 21 (84.0) | |
| **ER** | | | |
| Positive | 9 (14.5) | 53 (85.5) | **0.01** |
| Negative | 9 (42.9) | 12 (57.1) | |
| **PR** | | | |
| **Positive** | 7 (14.0) | 43 (86.0) | **0.04** |
| **Negative** | 11 (33.3) | 22 (66.7) | |
| **Tumor size** | | | |
| pT1-pT2 | 6 (20.0) | 24 (80.0) | 0.61 |
| pT3-pT4 | 12 (25.0) | 36 (75.0) | |
| **Histological grade** | | | |
| G1/G2 | 5 (14.7) | 29 (85.3) | 0.16 |
| G3 | 10 (28.6) | 25 (71.4) | |
| **Lymph node status** | | | |
| pN0-pN1 | 12 (24.5) | 37 (75.5) | 0.55 |
| pN2-pN3 | 5 (18.5) | 22 (81.5) | |

**Table 4. Association of EGFR and VEGF protein expressions with histopathological features.**

| Characteristics | VEGF | | |
|---|---|---|---|
| | Positive, N (%) | Negative, N (%) | p-Value |
| **Age** | | | |
| <50 | 11 (19.0) | 47 (81.0) | 0.19 |
| ≥50 | 8 (32.0) | 17 (68.0) | |
| **ER** | | | |
| Positive | 16 (25.8) | 46 (74.2) | 0.28 |
| Negative | 3 (14.3) | 18 (85.7) | |
| **PR** | | | |
| Positive | 11 (22.0) | 39 (78.0) | 0.75 |
| Negative | 8 (25.0) | 24 (75.0) | |
| **Tumor size** | | | |
| pT1-pT2 | 7 (22.6) | 24 (77.4) | 0.77 |
| pT3-pT4 | 12 (25.5) | 35 (74.5) | |
| **Histological grade** | | | |
| G1/G2 | 8 (22.9) | 27 (77.1) | 0.82 |
| G3 | 7 (20.6) | 27 (79.4) | |
| **Lymph node status** | | | |
| pN0-pN1 | 10 (20.4) | 39 (79.6) | 0.52 |
| pN2-pN3 | 7 (26.9) | 19 (73.1) | |

In this study, tumors with moderate or strong VEGF expressions were categorized as positive, while the rest were considered negative for comparison with other categorical variables. VEGF-A expression didn't show any significant association with age (p = 0.19), tumor size, grade, axillary lymph node, and hormone receptor statuses (Table 4).

## Discussion

To our knowledge, this study is the first of its kind to assess the EGFR and VEGF status among female invasive breast cancer patients in the Ethiopian population. Generally, studies have reported an overexpression of EGFR ranging from 16% to 36% in breast cancer cases [32]. Variation in EGFR positivity has been demonstrated in previous studies among different populations, including European (20.6%) [33], Tunisian (28.6%) [34], black American (25.2%) and White American (17.5%) [35]. It is important to note that different studies have reported varying results, even within the same country, due to the lack of standardized scoring systems [36–40]. Therefore, the implementation of a standardized protocol may be necessary for future Ethiopian breast cancer studies to accurately assess EGFR status. In our study, 21.7% of the breast tumor samples tested positive for EGFR

In the current study, a significant inverse association was observed between hormonal receptor positivity and EGFR expression, which is consistent with previous studies [35, 36, 41, 42]. This may be due to the down-regulation of hormone receptor expression by EGFR signaling [43–45]. EGFR has been indicated to be a potential drug target for triple-negative breast cancer cases [46]. On the other hand, ER is used as the primary therapeutic target for hormone receptor positive patients. However, EGFR overexpression has also been associated with resistance to endocrine treatment in this group [47]. Targeting EGFR in addition to endocrine treatment could potentially delay drug resistance and improve the survival rate of hormone receptor positive breast cancer cases [48].

Although EGFR positivity was more frequent in younger age cases (40% vs. 24%), the current study failed to show a significant association, which is in agreement with many studies [34, 41, 49–51]. Previous studies have reported a significant association between EGFR expression and younger age or premenopausal breast cancer cases [35, 41]. The variation in results compared to previous studies may be attributed to the small sample size, differences in the type of specimen the measurement technique for EGFR expression., type of antibodies used, different criteria for EGFR positivity or biological nature of the tumor. Our study also did not show a significant association with tumor size and lymph node status, which is consistent with previous studies [36, 41, 50–53]. However, a significant association of EGFR positivity with tumor size and lymph node status was reported in other studies [35, 54]. The variation in results compared to previous studies may be due to small sample size, differences in tissue processing, the type of assay used for EGFR measurement, the cut-off values for EGFR positivity, variation in breast cancer subtypes or ethnic disparities. The current study also showed a higher frequency of EGFR positivity among grade III tumors compared to grade I/II tumors, although this difference was not statistically significant. On the other hand, other studies have reported a significant association of EGFR over-expression with high-grade tumors in invasive types of breast carcinoma [34, 35, 51, 55]. The current study's findings indicate that there is a higher relative frequency of EGFR overexpression in high-grade tumors. This suggests that EGFR may be involved in the proliferation and invasion of breast cancer, both of which are characteristics commonly found in high-grade tumors [56]. The lack of significant association between EGFR expression and tumor grade in the current study may be attributed to differences in the selection of study groups, pre-analytical procedures, the type of primary antibodies used, the score used for EGFR interpretation, and the small sample size.

The majority of the tumors in the current study showed positive expression of VEGF-A (84.4%), which is similar to previous studies [57, 58]. The positivity rate of VEGF-A in this study was higher compared to other reports [59, 60]. However, Cimpean et al (2008) reported an increased frequency of VEGF positivity [61]. Dhakal et al (2012) also reported a higher frequency of moderate VEGF-A expression in Norwegians compared to the current study [31]. The variations in positivity rate may be attributed to differences in sample size, pre-analytical procedures, type of primary antibodies used for detection, scoring criteria for VEGF positivity or genetic polymorphism [62–66].

In the current study, no significant association was observed between expressions of VEGF and hormone receptors, which is consistent with several previous studies [60, 65, 67–69]. However, other studies have reported inverse correlations [70, 71]. The lack of association in our study may be attributed to the small sample size, differences in types of cases, scoring criteria for VEGF positivity, and variation in VEGF detection techniques. Co-expression of hormone receptors and VEGF-A has been shown to be associated with shorter recurrence-free and overall survival even after adjuvant endocrine treatment [71, 72].

Our study did not find a significant association between VEGF expression and age, which is consistent with previous studies [59, 60, 65, 67, 71, 73]. Whereas, a study of 574 tumors in cases of node-negative invasive breast cancer revealed a significant increase in levels of VEGF expression among older patients [74]. Furthermore, a consistent increase in levels of VEGF-A mRNA expression was observed in older age primary breast cancer cases [68]. VEGF expression was also shown to decrease with age in normal breast tissues but not in malignant breast specimens [75]. In the current study, the percentage of VEGF positivity was higher among older age groups, although it was not statistically significant. The lack of a significant association in the current study may be due to factors such as sample size, type of specimens, type of antibodies used, or the natural behavior of the tumor tissues. The present study did not show a significant association between VEGF-A expression and histopathological features (tumor

size, histologic grade and axillary lymph node status), which is consistent with previous studies [54, 65, 76]. However, other studies have reported a significant association with lymph node status [56, 67]. Furthermore, conflicting results have been reported regarding the association with tumor size [60, 71, 77–79] and grade [67, 71, 80, 81]. The variation in results between this study and previous studies may be attributed to differences in sample size, tissue fixation and processing, types of VEGF-A clones, scoring system for positivity and dichotomization of categorical variables. Additionally, the lack of association between VEGF expression and histopathological features in our study may also be due to the minimal role of VEGF-A in autocrine signaling as a result of reduced expression of VEGFR2 (the major VEGF-A receptor) by the tumor cells [82], or partly due to increased secretion of other angiogenic factors such as fibroblast growth factor, which contributes to tumor growth [83]. Although the expression of VEGFRs by breast tumor cells is controversial, studies have found low expression of VEGFR2 by breast tumor cells [84–86].

The current study has some limitations. The first limitation is the small sample size. These limitations may restrict the ability of the current study to draw definitive conclusions. Secondly, because the status of HER-2 expressions is unknown, the frequency of TNBC and its association with VEGF-A and EGFR could not be determined. Thirdly, this study did not include prospective or retrospective observations. Therefore, the prognostic values of the study parameters were not assessed.

## Conclusion

In the current study, there was a significant association between EGFR expression and hormone receptor status. This suggests that EGFR could be a reliable predictor for adjuvant therapy in cases of hormone receptor negative breast cancer. Although it is difficult to draw definitive conclusion due to other unstudied parameters, EGFR might be one of the potential therapeutic targets for this group. Moreover, ER-negative patients who overexpress EGFR may benefit from anti-EGFR therapies if they are combined with other targeted drugs. Despite its limitations, this study lays the groundwork for future breast cancer studies in the Ethiopian population. Therefore, we recommend conducting further research with a sufficient sample size, comprehensive clinical information and well-designed methodologies.

## Acknowledgments

We would like to express our gratitude to the University of Michigan African Studies Center, the University of Michigan African Presidential Scholars (UMAPS) program, the University of Michigan Comprehensive Cancer Center, and Addis Ababa University for their technical and material support. We also extend our thanks to Professor Senait Fisseha from the University of Michigan for her generous assistance in obtaining a funding and ensuring the practicality of this work.

## Author Contributions

**Conceptualization:** Sisay Addisu, Abebe Bekele.

**Data curation:** Sisay Addisu, Mark J. Hoenerhoff.

**Formal analysis:** Sisay Addisu, Mark J. Hoenerhoff, Sofia D. Merajver.

**Funding acquisition:** Sisay Addisu.

**Investigation:** Sisay Addisu.

**Methodology:** Sisay Addisu, Abebe Bekele, Mathewos Assefa, Tufa Gemechu, Sofia D. Merajver.

**Software:** Sisay Addisu.

**Supervision:** Sisay Addisu, Abebe Bekele, Daniel Seifu, Mathewos Assefa, Tufa Gemechu, Mark J. Hoenerhoff, Sofia D. Merajver.

**Visualization:** Mark J. Hoenerhoff.

**Writing – original draft:** Sisay Addisu.

**Writing – review & editing:** Abebe Bekele, Mathewos Assefa, Tufa Gemechu, Sofia D. Merajver.

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
