## [Decision Letter · Decision Letter 0]

6 Sep 2023

PONE-D-23-18869EGFR and VEGF-A Expressions in Ethiopian Female Breast Cancer and Their Association with Histopathologic FeaturesPLOS ONE

Dear Dr. Bekele,

Thank you for submitting your manuscript to PLOS ONE. After careful consideration, we feel that it has merit but does not fully meet PLOS ONE’s publication criteria as it currently stands. Therefore, we invite you to submit a revised version of the manuscript that addresses the points raised during the review process.

Please submit your revised manuscript by Oct 21 2023 11:59PM. If you will need more time than this to complete your revisions, please reply to this message or contact the journal office at plosone@plos.org. Please include the following items when submitting your revised manuscript:A rebuttal letter that responds to each point raised by the academic editor and reviewer(s). You should upload this letter as a separate file labeled 'Response to Reviewers'.A marked-up copy of your manuscript that highlights changes made to the original version. You should upload this as a separate file labeled 'Revised Manuscript with Track Changes'.An unmarked version of your revised paper without tracked changes. You should upload this as a separate file labeled 'Manuscript'.

We look forward to receiving your revised manuscript.

Kind regards,

Alessandro Rizzo

Academic Editor

PLOS ONE

Journal Requirements:

a) The name of the colleague or the details of the professional service that edited your manuscript.

b) A copy of your manuscript showing your changes by either highlighting them or using track changes (uploaded as a *supporting information* file).

c) A clean copy of the edited manuscript (uploaded as the new *manuscript* file).

http://etd.aau.edu.et/handle/123456789/21346

In your revision ensure you cite all your sources (including your own works), and quote or rephrase any duplicated text outside the methods section. Further consideration is dependent on these concerns being addressed.

Reviewers' comments:

Reviewer's Responses to Questions

**Comments to the Author**

1. Is the manuscript technically sound, and do the data support the conclusions?

Reviewer #1: Partly

Reviewer #2: No

Reviewer #3: Yes

2. Has the statistical analysis been performed appropriately and rigorously? 

Reviewer #1: Yes

Reviewer #2: Yes

Reviewer #3: N/A

3. Have the authors made all data underlying the findings in their manuscript fully available?

Reviewer #1: Yes

Reviewer #2: Yes

Reviewer #3: Yes

4. Is the manuscript presented in an intelligible fashion and written in standard English?

Reviewer #1: No

Reviewer #2: No

Reviewer #3: Yes

5. Review Comments to the Author

Reviewer #1: The study assesses a current, timely topic in breast cancer.

We recommend some changes:

- We believe this article is suitable for publication in the journal although major revisions are needed. The main strengths of this paper are that it addresses an interesting and very timely question and provides a clear answer, with some limitations. Certainly, the authors should better highlight the limitations of the current paper, including the small sample size. In addition, the samples are quite old.

- A linguistic revision is needed.

- The background of the changing scenario of medical treatment in breast cancer should be better discussed, and some recent papers regarding this topic should be included ( PMID: 35171746 ; PMID: 36633661; PMID: 34802383; PMID: 34793275 ).

Major changes are necessary.

Reviewer #2: The respected authors aimed to investigate the "EGFR and VEGF-A Expressions in Ethiopian Female Breast Cancer and Their Association with Histopathologic Features " .The study is well-designed, however ,this study lacks novelty and the references are not up to date. They are mostly outdated. The reasons for conducting this study have not been well presented with good documentation

Reviewer #3: Authors present the results of a study assessing the correlation between the expression of two breast cancer biomarkers (EGFR and VEGF) and their correlation with hormone receptor status and histopathological characteristics invasive type of female breast cancer among a specific ethnical group (Ethiopians).

Since the focus on a minority/underrepresented group is what makes this paper particular, more effort should be given to explain which selected treatment strategies could be implemented in this setting, in addition to "EGFR being a good predictor in adjuvant therapy for hormone receptor negative breast cancer cases" which has been already shown in other ethnic groups.

Further explaination should be given on why EGFR and VEGF expressions didn’t show significant correlation with tumor size, grade, lymph node status and age at diagnosis. If inadequate sample size ends up being the major accountable consider increasing it.

VEGF has been described to be increased in invasive breast cancer and related to lower progression free survival: further explanation should be given to explain the opposite results in this study on the former statement and assessment of the latter should be taken into account.

A TNBC sub-group should be considered.

6. PLOS authors have the option to publish the peer review history of their article (what does this mean?). If published, this will include your full peer review and any attached files.

Reviewer #1: No

Reviewer #2: No

Reviewer #3: No

---

## [Author Response · Author response to Decision Letter 0]

8 Nov 2023

Dear

Here, I attached the rebuttal letter, the manuscript with marked-up copy and unmarked revised manuscript.

---

## [Decision Letter · Decision Letter 1]

29 Mar 2024

PONE-D-23-18869R1EGFR and VEGF-A Expressions in Ethiopian Female Breast Cancer and Their Association with Histopathologic FeaturesPLOS ONE

Dear Dr. Bekele,

Thank you for submitting your manuscript to PLOS ONE. After careful consideration, we feel that it has been significantly improved after revision iut does not fully meet PLOS ONE’s publication criteria as it currently stands. Therefore, we invite you to submit a revised version of the manuscript that addresses the points raised during the review process.

We look forward to receiving your revised manuscript.

Kind regards,

Elda Tagliabue

Academic Editor

PLOS ONE

Reviewers' comments:

Reviewer's Responses to Questions

**Comments to the Author**

1. If the authors have adequately addressed your comments raised in a previous round of review and you feel that this manuscript is now acceptable for publication, you may indicate that here to bypass the “Comments to the Author” section, enter your conflict of interest statement in the “Confidential to Editor” section, and submit your "Accept" recommendation.

Reviewer #3: All comments have been addressed

Reviewer #4: (No Response)

2. Is the manuscript technically sound, and do the data support the conclusions?

Reviewer #3: Yes

Reviewer #4: Yes

3. Has the statistical analysis been performed appropriately and rigorously? 

Reviewer #3: N/A

Reviewer #4: Yes

4. Have the authors made all data underlying the findings in their manuscript fully available?

Reviewer #3: Yes

Reviewer #4: Yes

5. Is the manuscript presented in an intelligible fashion and written in standard English?

Reviewer #3: Yes

Reviewer #4: Yes

6. Review Comments to the Author

Reviewer #3: Authors have revised the paper to the best of its potential.

Although some important issues are still unmet, such as correlation between tumor size, grade, lymph node status and age at diagnosis and EGFR/VEGF expressions and inclusion of a TNBC sub-group, acceptable explenations have been given to clarify the others.

Reviewer #4: General comment!

1. The method section should be clear, specially, IHC scoring, inclusion and exclusion criteria?

2. Some percentage disagreement in the result should be revised

3. Better to indicate the scientific and clinical value of these markers rather than simple comparison with other results

All necessary comments are uploaded in the word document.

7. PLOS authors have the option to publish the peer review history of their article (what does this mean?). If published, this will include your full peer review and any attached files.

Reviewer #3: No

Reviewer #4: **Yes: **Esmael Besufikad Belachew

---

## [Author Response · Author response to Decision Letter 1]

7 Jul 2024

General comments of reviewer 4

Comment 1: The method section should be clear, specially, IHC scoring, inclusion and exclusion criteria?

Response 1: Within the manuscript, we tried to correct the methodology and clarified it based on the comments given.

Comment 2: Some percentage disagreement in the result should be revised. 

Response 2: The disagreement arose due to errors in writing the data, which have now been included and corrected. 

Comment 3: Better to indicate the scientific and clinical value of these markers rather than simple comparison with other results. All necessary comments are uploaded in the word document.

Response 3: As much as we can, apart from comparison, we added some clinical values of these markers. We corrected based on the comments given in the word document.

---

## [Decision Letter · Decision Letter 2]

24 Jul 2024

Epidermal growth factor receptor (EGFR) and vascular endothelial growth factor A (VEGF-A) expressions in Ethiopian female breast cancer and their association with histopathologic features

PONE-D-23-18869R2

Dear Dr. Bekele,

We’re pleased to inform you that your manuscript has been judged scientifically suitable for publication and will be formally accepted for publication once it meets all outstanding technical requirements.

Kind regards,

Daniele Ugo Tari, M.D.

Academic Editor

PLOS ONE

Additional Editor Comments (optional):

Reviewers' comments:

Reviewer's Responses to Questions

**Comments to the Author**

1. If the authors have adequately addressed your comments raised in a previous round of review and you feel that this manuscript is now acceptable for publication, you may indicate that here to bypass the “Comments to the Author” section, enter your conflict of interest statement in the “Confidential to Editor” section, and submit your "Accept" recommendation.

Reviewer #4: All comments have been addressed

2. Is the manuscript technically sound, and do the data support the conclusions?

Reviewer #4: Partly

3. Has the statistical analysis been performed appropriately and rigorously? 

Reviewer #4: Yes

4. Have the authors made all data underlying the findings in their manuscript fully available?

Reviewer #4: Yes

5. Is the manuscript presented in an intelligible fashion and written in standard English?

Reviewer #4: No

6. Review Comments to the Author

Reviewer #4: Still needs improvement in writing

The Ethical approval section in the declaration section is enough, please remove the first one

7. PLOS authors have the option to publish the peer review history of their article (what does this mean?). If published, this will include your full peer review and any attached files.

Reviewer #4: **Yes: **Esmael Besufikad Belachew

---

## [Editor Report · Acceptance letter]

29 Aug 2024

PONE-D-23-18869R2 

PLOS ONE

Dear Dr. Addisu, 

I'm pleased to inform you that your manuscript has been deemed suitable for publication in PLOS ONE. Congratulations! Your manuscript is now being handed over to our production team.

Kind regards, 

on behalf of

Dr. Daniele Ugo Tari 

Academic Editor

PLOS ONE